# Persistent Organic Pollutant-Mediated Insulin Resistance

**DOI:** 10.3390/ijerph16030448

**Published:** 2019-02-03

**Authors:** Yeon A. Kim, Joon Beom Park, Min Seok Woo, Sang Yeob Lee, Hye Young Kim, Young Hyun Yoo

**Affiliations:** 1Department of Anatomy and Cell Biology and Mitochondria Hub Regulation Center, Dong-A University College of Medicine, Busan 49201, Korea; keivin@naver.com (Y.A.K.); csplen1990@dau.ac.kr (J.B.P.); leesy@dau.ac.kr (S.Y.L.); dolph02@dau.ac.kr (H.Y.K.); 2Department of Anesthesiology and Pain Medicine, Gyeongsang National University Changwon Hospital, Changwon 51472, Korea; 3Institute of Health Sciences, School of Medicine, Gyeongsang National University, Jinju 52727, Korea; 4Department of Convergence Medical Science, Gyeongsang National University, Jinju 52727, Korea; whitewms@naver.com; 5Department of Rheumatology, Dong-A University College of Medicine, Busan 49201, Korea

**Keywords:** insulin resistance, persistent organic pollutants

## Abstract

Persistent organic pollutants (POPs) such as organochlorine (OC) pesticides, polychlorinated biphenyls (PCBs), polychlorinated dibenzo-p-dioxins (PCDDs), and polychlorinated dibenzofurans (PCDFs) have become wide-spread environmental contaminants as a consequence of their extensive use, long-range transport, and persistence. Because POPs are highly resistant to metabolic degradation, humans bioaccumulate these lipophilic and hydrophobic pollutants in fatty tissues for many years. Previous studies have demonstrated that POPs including PCBs are involved in the development of diabetes mellitus (DM) type 2 and insulin resistance. Numerous epidemiological studies suggest an association between POP burden and DM type 2/metabolic syndrome. In addition, several experimental studies have provided additional evidence supporting the association between POP exposure and DM type 2 or insulin resistance. Epidemiological and experimental studies have provided compelling evidence indicating that exposure to POPs increases the risk of developing insulin resistance and metabolic disorders. However, the detailed molecular mechanism underlying POP-induced insulin resistance is yet to be elucidated. In this article, we review literature that has reported on the association between POP burden and insulin resistance and the mechanism underlying POP-induced insulin resistance, and discuss implications for public health.

## 1. Persistent Organic Pollutants (POPs)

Numerous chemicals produced by humans intentionally or unintentionally have been released into ecosystems since the Industrial Revolution, and the propensity for these toxic chemicals to not degrade has been reported consistently since the 20th century. The most representative chemicals showing a well-known causal relationship are POPs [1].

POPs contain two basic groups of synthetic organic compounds: polycyclic aromatic hydrocarbons and halogenated hydrocarbons, which include several organochlorines (OCs), namely dioxin, furan, polychlorinated biphenyls (PCBs), Mirex, toxaphene, heptachlor, chlordane, and dichloro-diphenyl-trichloroethane (DDT). Historically, halogenated hydrocarbons have been shown to be the most resistant to degradation by photodegradation or heat, and halogenated hydrocarbons possess low solubility in water, high solubility in lipids, and are global produced, used, and released. OCs are typically the most persistent of all halogenated hydrocarbons. Universally, the more highly chlorinated biphenyls tend to accumulate to a greater extent than the less chlorinated PCBs; likewise, metabolism and excretion are slower for the highly chlorinated biphenyls than for the less chlorinated PCBs. Polychlorinated dibenzo-p-dioxins (PCDDs) and polychlorinated dibenzofurans (PCDFs) are polyhalogenated aromatic hydrocarbons that exert high toxicity. There are 210 different congeners, including 75 dioxin congeners and 135 furan congeners, of which 17 are potentially toxic. PCBs are a family of 209 congeners for which there are no known natural sources [2].

These substances accumulate with the highest concentration in humans, a top species in the food chain, while remaining in the soil or water and exhibiting toxicity for many years [3]. In May 2001, 12 POP species (aldrin, chlordane, DDT, dieldrin, endrin, heptachlor, hexachlorobenzene (HCB), Mirex, toxaphene, PCBs, PCDDs, and PCDFs) were initially prohibited for use through the Stockholm Convention. Thereafter, nine additional POPs were prohibited in May 2009 [4].

Generally, POPs not only have long half-lives, persisting in the environment for years or decades, but also are widely dispersed around the world, through the air, water currents, and living organisms. Additionally, POPs bioaccumulate and biomagnify, penetrating the food chain; in other words, they bioconcentrate at higher levels in food webs, thus polluting and exposing all living things, including humans. POPs are linked with serious health risks in humans and other living organisms [5]. Since most POPs are lipophilic, they tend to remain in fat-rich tissues, such as adipose tissues.

POPs have been measured in various living organisms [6]. POPs have been measured in biological samples, such as human blood, body fat, and breast milk in studies around the world. These chemicals are not well metabolized or excreted. Thus, even small doses that are ingested daily can accumulate to yield detectable amounts over time [3].

## 2. Human Implications of POPs

The awareness of POPs was the result of large-scale casualties caused by the exposure to high concentrations of POPs in the early 20th century. From 1956 to 1961, more than 4000 cases of porphyria occurred in Eastern Turkey due to the ingestion of HCB [7]. In 1968 and 1979, Japan and Taiwan, respectively, consumed approximately 1200 types of contaminated cooking oil, resulting in “Yusho” (oil disease), which includes symptoms, such as reproductive dysfunction, severe chloracne, hyperpigmentation, discharge from eye, headaches, vomiting, fever, visual disturbances, and respiratory problems. Between 1962 and 1971, there was a strong positive relationship between developmental soft tissue sarcoma, non-Hodgkin’s lymphoma, Hodgkin’s disease, chloracne, and chronic lymphocytic leukemia, and the use of Agent Orange including 2,3,7,8-tetrachlorodibenzo-p-dioxin (TCDD) [8]. It has also been reported that diabetes mellitus (DM) type 2, hypertension, heart disease, and chronic respiratory conditions are associated with POP exposure [9].

Likewise, POPs are toxic at high levels as demonstrated by mass poisoning incidents. Exposure to high levels of POPs is associated with serious human health problems including death, disease, and birth defects among humans and animals [10]. Specific health risks can include cancer, allergies, hypersensitivity, and damage to the immune, neurological, and reproductive systems.

Initially, it was difficult to reveal the causal relationship between the effects of chronic POPs exposure at very low concentrations compared with high concentrations of acute toxicity, which was revealed much later. According to *State of the Science of Endocrine Disrupting Chemicals—2012* published by the WHO in 2012 [1], chronic exposure of lower concentrations of POPs has been shown to lead to female reproductive dysfunction [11,12], testicular cancer [13,14], breast cancer [15], prostate cancer [16], decreased semen quality [17,18,19], increased cryptorchidism and hypospadias at birth [20,21], and cognitive and behavioral deficits caused by developmental exposure [22,23,24]. In particular, these neurodevelopmental disorders have been linked to severe forms of thyroid hormone deficiencies, and the decrease of thyroid function has been associated with PCBs, PBDEs, phthalates, bisphenol A, and perfluorinated chemicals in some epidemiological studies [25]. Thyroid cancer has also been to have a weak association with pesticides and 2,3,7,8-tetrachlorodibenzo-p-dioxin. With regard to metabolic disorders, decreased bone mineral density or increased risk of bone fractures, obesity, DM type 2, and metabolic syndrome due to the disruption of the energy storage–energy balance endocrine system are suspected to be potentially sensitive to endocrine-disrupting chemicals (EDCs) [26,27,28,29].

OCs, PCBs, HCB, and pesticides, including DDT and lindane (g-hexachlorocyclohexane, HCH), are classified as latent carcinogens to humans according to the International Agency for Research on Cancer [30,31]. Regarding the mechanism of cancer development by POPs, an epigenetic mechanism (i.e., chromosomal instability, abnormal gene expression, and DNA methylation) has been suggested, and the inverse relationship between DNA global methylation levels and blood plasma levels for several POPs has been reported [32]. Furthermore, there are strong epidemiological research data associating the exposure to phthalates with airway disorders, including asthma, and the exposure to phthalates and dioxins is associated with endometriosis and allergies.

PCBs that are intensively being released into the environment are also carcinogenic in nature, because they are weakly estrogenic, and some OCs have been tested almost exclusively in epidemiological studies in breast [33,34], prostate [34], colorectal, and endometrial cancers [35], and in non-Hodgkin’s lymphoma [36]. Furthermore, the neurotoxic of effects of PCBs are related to dose–response and structure–activity relationships (SAR). Sufficient epidemiological and experimental evidence has shown that PCB exposure is associated with motor and cognitive deficits in humans and animal models [37]. 

## 3. POPs as ECDs

Endocrine-disrupting chemicals (EDCs) disturb the immune, reproductive, and nervous system in humans and animals. Several studies have previously indicated that POPs are ECDs [38,39]. Recently, many epidemiological studies have provided evidence regarding the relationship between POPs and metabolic disorders. Lee et al. revealed that the levels of Methanobacteriales in the human gut were associated with higher body weight and waist circumference [40]. Janesick et al. suggested the EDCs as an obesogene [41]. The epidemiologic evidences for association between DM type 2 and EDCs were little by little accumulated recently [42,43,44,45,46]. Although rare, the causality of DM type 1 by EDCs also reported [47]. Metabolic syndrome (which is defined clinically as hypertension, abdominal (central) adiposity, increased serum triglycerides, low serum high density lipoproteins (HDL), and high blood sugar, even after fasting [48]) is an important disease group that can cause obesity and DM type 2 [49]. Although several studies also reported the progression of the metabolic syndrome, obesity, and DM type 2 by EDCs [50,51,52], experimental data revealing the mechanism underlying POPs exposure-induced endocrine disruption are lacking. 

An elaborate study revealed that PCB-77 may contribute to the development of obesity and obesity-associated atherosclerosis [53]. The study that examined the in vitro and in vivo effects of PCB-77 and TCDD demonstrated that low concentrations of PCB-77 or TCDD increased adipocyte differentiation, glycerol-3-phosphate dehydrogenase activity, and the expression of peroxisome proliferator-activated receptor gamma. In addition, PCB-77 was shown to promote the expression and release of various proinflammatory cytokines in vitro, and PCB-77 resulted in an increase in body weight, adipocyte hypertrophy, serum dyslipidemia, and augmented atherosclerosis in vivo. However, this study did not address insulin resistance.

## 4. Epidemiologic Evidence of POP-Induced Insulin Resistance

The relationship between POPs and DM type 2 or insulin resistance was not an important issue in the early 20th century, although it has been suggested for people who were constantly and chronically exposed to low concentrations of POPs in the 1990s. The studies regarding exposure to POPs, including TCDD, which is the most potent congener of dioxin, or other POPs in occupational or accidental settings, have reported an increased risk of DM type 2, modified glucose metabolism, and insulin resistance [54,55,56,57,58].

The issue of the fully encompassing problem has been reported as a strong relationship between the serum concentration of six POPs (2,2,4,4,5,5- hexachlorobiphenyl, 1,2,3,4,6,7,8-heptachlorodibenzo-p-dioxin, 1,2,3,4,6,7,8,9-octachlorodibenzo-p-dioxin, oxychlordane, p,p-dichlorodiphenyltrichloroethane, and trans-nonachlor) and the prevalence of DM type 2 [46]. This cross-sectional prospective study involved 2,016 adult participants, and the association was strong despite the adjustment of several confounding factors and stratified analyses, and OCs or nondioxin-like PCB (PCB-153) were found to be most strongly correlated with DM type 2 notably. Furthermore, these authors investigated the relationship between serum concentrations of POPs, especially OC pesticides or nondioxin-like PCBs, and insulin resistance, pre-stage diabetes mellitus, or the potential risk of DM type 2 in nondiabetic adults [59]. This study also showed the association between POPs and insulin resistance, and the authors even suggested the possibility of interaction with obesity to increase the risk of DM type 2. Although the sample size was small, a case–control study has also been reported. For 50 nondiabetic subjects with metabolic syndrome and 50 normal controls, the association between eight OC pesticides and metabolic syndrome was examined and only heptachlor epoxide was related meaningfully [60].

Beyond the simple examination of the association between POPs and DM type 2, the predictive potential of the occurrence of DM type 2 from POP levels in serum has been examined. Lee et al. measured the serum levels of 8 OC pesticides, 22 PCB congeners, and 1 polybrominated biphenyl (PBB) of 90 controls subjected that remained free of DM type 2. The 90 cases developed DM type 2 in 1987~1988 and in 2005~2006. Although the serum levels of POPs were very low, these levels were very similar to exposure levels observed in nature which increased the risk of DM type 2, suggesting an important role for POPs in current trends in DM type 2 due to obesity [61].

Recently, a toxicology program workshop thoroughly reviewed 72 published epidemiological studies that investigated the associations of POPs with DM type 2 prior to the assessment [62]. According to the review, the association between DM type 2 and OC compounds such as *trans*-nonachlor, dichlorodiphenyldichloroethylene (DDE), PCBs, dioxins, and dioxin-like chemicals were found to be strongly correlated; however, associations between other non-OC POPs, such as perfluoroalkyl acids and brominated compounds, and DM type 2 were found to be less correlated. However, as the review study also indicated, further experimental data are required to support the epidemiological studies.

## 5. Current Concepts Regarding the Mechanism Underlying Insulin Resistance

Insulin resistance can occur through dysfunction of insulin signaling pathway. Current concepts of insulin signaling pathways are depicted in Figure 1.

### 5.1. Mechanism of Insulin Resistance

Insulin resistance, a condition where cellular responses to insulin are unsuitable, is found primarily in insulin-sensitive tissues, liver, muscle and fat. Insulin resistance can result from various situations including abnormal insulin signaling, lipotoxicity, inflammation, mitochondrial dysfunction, and endoplasmic reticulum (ER) stress. These mechanisms are chiefly mediated by inhibitory serine/threonine phosphorylation, dephosphorylation, transcriptional modifications, posttranslational modifications, and genetic mutations [63].

### 5.2. Insulin Receptor

Insulin signaling is initiated through insulin binding with the extracellular domains of the insulin receptor, followed by receptor autophosphorylation of several tyrosine residues located in intracellular domains. This tyrosine residue interacts with various adaptor proteins, including insulin receptor substrate-1 and -2 (IRS-1 and IRS-2, respectively) and SH2 domain-containing protein (SHC), which bind to intracellular receptor sites and become phosphorylated. Mutations on the insulin receptor have been identified in several conditions such as leprechaunism, Rabson-Mendenhall syndrome, or the type-A syndrome of insulin resistance, but have not been observed in patients with typical DM type 2 [63].

### 5.3. IRS Protein

IRS-1 and IRS-2 recruit the formation of molecular complexes and activate downstream signaling cascades. For example, in IRS-1-deficient mice, insulin resistance develops mainly due to decreased insulin-stimulated glucose metabolism in the muscle alone. In IRS-2-deficient mice, however, multiple defects impact the liver, muscle and adipose tissue, which includes reduced peripheral glucose utilization, reduced suppression of endogenous glucose production, and reduced hepatic glycogen synthesis [64].

### 5.4. PI 3-kinase/Akt Signaling

The key target of the IRS protein is phosphoinositide 3-kinase (PI 3-kinase). PI3-kinase activates phosphoinositide-dependent kinase-1 (PDK1 and PDK2) by the phosphorylation of phosphatidylinositol 4,5 bisphosphate (PIP2) into phosphatidylinositol 3,4,5 triphosphate (PIP3) [65]. Activated PDK1 phosphorylates serine/threonine kinases including Akt/protein kinase B (PKB) and atypical protein kinase C λ and ζ (PKCλ/ζ) [66,67]. Akt kinase plays a central and varying role in biological processes, including cell growth, survival and, metabolism and responses to hormones, growth factors, and cytokines in numerous cell types [68]. In an in vivo study, Akt2-deficient mice showed insulin resistance in the liver, skeletal muscle, and adipose tissue [69,70]. Among three different isoforms of Akt (Akt 1, 2, and 3), Akt2 is chiefly found in insulin-responsive metabolic tissues and is essential for insulin metabolic processes. Insulin-stimulated Akt2 results in the uptake of circulating glucose through GLUT4 translocation from intracellular compartments to the cell membrane, especially in skeletal muscle and adipose tissue [71]. Insulin signaling to Akt in the liver is crucial to the suppression of glucose production and increased lipid synthesis [72,73]. Insulin–Akt signaling inhibits FoxO1 as a transcription factor that enhances the expression of the gluconeogenic enzymes phosphoenolpyruvate carboxykinase (PEPCK) and glucose 6-phosphatase (G6Pase) [74]. Akt phosphorylation also induces the activation of the SREBP1c transcription factor, which leads to de novo lipid synthesis [73].

### 5.5. Roles of Lipotoxicity in Insulin Resistance

The plasma free fatty acid (FFA) level is controlled by insulin. When the FFA level is continuously elevated, it causes lipotoxicity in non-adipose tissues and insulin resistance, and DM type 2 can occur [75]. High levels of plasma FFA induce the activation of c-Jun N-terminal kinase (JNK), IκB kinase (IKK), and PKC and IRS-1 Ser-307 phosphorylation [76]. Among the FFAs, palmitate especially promotes insulin resistance by ER stress, cytokine production, and activating JNK, and NF-κB [77,78,79]. Diacylglycerol (DAG) and ceramide, which are intermediate metabolites of FFAs, also induce insulin resistance. Increased muscle DAG causes insulin resistance by activating PKC-θ and IRS-1 Ser-307 phosphorylation [80]. Ceramide activates PKC and JNK, inhibits Akt activation via Akt Thr-34 phosphorylation, and increases the interaction of protein phosphatase 2 (PP2A) with Akt [81,82,83,84].

### 5.6. Roles of Inflammation in Insulin Resistance

In obesity, chronic and low-grade inflammation prevails and is involved in the pathogenesis of various chronic diseases. Pro-inflammatory cytokines secreted in the adipose tissue and by macrophages such as tumor necrosis factor alpha (TNF-α), interleukin-1β (IL-1β), and IL-6 can promote insulin resistance by multiple mechanisms [85,86], which include Ser/Thr kinase activation and decreases in IRS-1, glucose transporter type 4 (GLUT-4), and peroxisome proliferator-activated receptor gamma (PPARγ) expression or suppressor of cytokine signaling 3 (SOCS-3) activation [63,87,88,89]. The activation of Toll-like receptors (TLRs), especially TLR-2 and TLR-4, is also an important factor in inflammation-associated insulin resistance [78,90].

### 5.7. Roles of Mitochondrial Dysfunction in Insulin Resistance

The level of reactive oxygen species (ROS) as a byproduct of the electron transport chain in mitochondria is increased as a result of insulin resistance [91]. Increased ROS levels can be caused by reduced antioxidant enzymes, and alterations in mitochondrial proteins [92,93]. Increased ROS levels activate the phosphorylation of the insulin receptor and the insulin receptor substrate and decrease the level of the FoxO1 transcriptional factor, which eventually results in insulin resistance [93,94]. Because FFA metabolism is chiefly mediated by mitochondria, a decrease in mitochondrial function can lead to FFA and lipid accumulation and subsequent insulin resistance.

### 5.8. Roles of ER Stress in Insulin Resistance

The endoplasmic reticulum (ER) has many functions including protein folding, posttranslational modifications, and calcium storage. Some physiological conditions that increase the demand for protein folding or the stimuli that disrupt protein folding, result in the accumulation of unfolded or misfolded proteins in the ER lumen [95]. For re-establishing the protein-folding capacity and preventing the accumulation of unfolded or misfolded proteins, a mechanism known as the unfolding protein response (UPR) is activated by the alteration of transcriptional and translational processes [95]. Three ER membrane-associated proteins, which include PKR-like eukaryotic initiation factor 2α kinase (PERK), inositol requiring enzyme1 (IRE1), and activating transcription factor-6 (ATF6), are known as ER membrane-associated proteins to be related to the UPR and activate inflammatory pathways, such as TNF-α, IL-1β, and IL-6 by nuclear factor kappa-light-chain-enhancer of activated B cells (NF-κB) activation and finally result in insulin resistance [96]. In the case of IRE kinase, the phosphorylation of IRE-1α recruits the tumor necrosis factor receptor-associated factor 2 (TRAF2) protein, which activates JNK and subsequent IRS-1 phosphorylation [79].

## 6. Experimental Animal Studies Investigating the Mechanism Underlying POP-Induced Insulin Resistance

Several previous experimental studies have provided evidence supporting the association between POPs exposure and DM type 2 or insulin resistance [97,98,99,100]. Ruzzin et al. demonstrated that chronic exposure to low doses of a POP mixture, which is commonly found in food chains, induced the severe impairment of whole-body insulin activity and contributed to the development of abdominal obesity in rats. The in vitro treatment of differentiated adipocytes with nanomolar concentrations of POPs mixtures, which mimic those found in crude salmon oil, induced a significant inhibition of insulin-dependent glucose uptake [97]. A causal relationship between POPs and insulin resistance was demonstrated by subsequent studies. Ibrahim et al. elucidated that the chronic consumption of farmed salmon containing POPs causes insulin resistance and obesity in mice [100]. Gray et al. also provided evidence supporting that chronic exposure to PCBs (Aroclor 1254) exacerbates obesity-induced insulin resistance and hyperinsulinemia in mice [99]. Lv et al. studied the consequences of gestational and lactational exposure to a POP perfluorooctane sulfonate (PFOS) on the effects of pre-DM in offspring. In the study, they demonstrated that glucose and lipid homeostasis in adult rats is impaired by early-life exposure to PFOS [98]. These experimental studies provide compelling evidence that exposure to POPs increases the risk of developing insulin resistance and metabolic disorders. However, to date, the detailed molecular mechanism underlying POP-induced insulin resistance is yet to be elucidated.

The lack of information on the molecular aspects of POP-induced insulin resistance might primarily be a result of the lack of interest in this issue by contemporary life scientists who focus on cell and molecular biology. Difficult experimental approaches might also limit further study. Although epidemiological findings suggest an association between low-level chronic exposure to certain POPs and disease outcomes, the chronic exposure to POPs at low doses requires a long period of time to investigate. Therefore, many studies have investigated single POP treatments for a short duration. Thus, it is not completely clear that the data obtained from these experimental systems are relevant to human exposure. However, the exposure to multiple contaminants found in the environment could produce increased adverse effects by synergistic toxicity mechanisms, resulting in the lack of information [101].

Despite this limitation, numerous experimental studies have provided data supporting the association between POPs exposure and DM type 2. Experimental studies reporting that POPs alter glucose transport activity [102,103,104,105] and that POPs increase adiposity [85,106,107] have provided insight regarding the mechanism underlying POP-induced insulin resistance because increased adiposity as well as altered glucose transport activity are associated with insulin resistance. However, these studies did not reveal the detailed mechanism by which POPs alter insulin signaling. An experimental study has shown that chronic consumption of salmon containing POPs impairs the ability of insulin to stimulate Akt phosphorylation in vivo. Another study has shown that PCB-153 impairs insulin sensitivity through the dysregulation of hepatocyte nuclear factor 1 b (HNF1b)/ROS/NF-κB [108]. Recently, we revealed the mechanism underlying the insulin resistance induced by PCB-118 and PCB-138 [109]. In that study, we demonstrated that PCB-118 or PCB-138 promotes large lipid droplet (LD) formation through fat-specific protein 27 (Fsp27). In addition, we elucidated that PCB-118 or PCB-138 impair the insulin-induced upregulation of p-Akt (Ser473) and p-PI3K p85 (Tyr458). Importantly, we elicited that Fsp27 mediates PCB-induced insulin resistance via IRS1 downregulation [109]. In another study, we demonstrated the mechanism underlying the obesity induced by PCB-138. In that study, we elucidated that LD enlargement induced by PCB-138 confers adipocytes the resistance to TNF-α-induced cell death. In addition, we elicited that Fsp27, perilipin, and survivin, at least in part, are involved in sustaining enlarged LDs, which contributes to the induction of obesity and subsequent insulin resistance [110].

## 7. Future Tasks

According to previous epidemiological studies, the overall evidence is sufficient for a positive association of some OC POPs with DM type 2. However, further experimental data are needed to confirm the causality of these POPs. Dynamic experimental studies might not only prove the causality but also provide important insights into the pathogenesis of DM type 2 and the mechanisms governing POP-mediated insulin resistance. Experimental studies could offer a suitable target for interventions targeting PCB-induced insulin resistance. For example, a previous study demonstrated that PCB-induced impairment of glucose homeostasis in mice can be prevented by resveratrol, potentially through the stimulation of Nrf2 signaling and enhanced insulin-stimulated glucose disposal in adipose tissue [111]. In a previous study, we showed that the depletion of the Fsp27 gene resulted in the inhibition of LD enlargement and attenuation of insulin resistance [109]. Furthermore, we demonstrated that metformin, a representative insulin resistance-improving drug, alleviates PCBs-induced insulin resistance through Fsp27. These reports, in conjunction with future studies, could provide us with an avenue for interventions targeting PCBs-induced insulin resistance.

## 8. Conclusions

Numerous epidemiological studies have provided compelling evidence indicating that exposure to POPs increases the risk of developing insulin resistance and metabolic disorders. Several experimental studies have provided evidence supporting the association between POP exposure and DM type 2 or insulin resistance. However, the detailed molecular mechanism underlying POP-induced insulin resistance is yet to be elucidated. Despite these limitations of experimental approaches, experimental studies which have been recently reported provided new insight regarding the pathogenesis of DM type 2. Experimental studies could offer a suitable target for interventions targeting PCB-induced insulin resistance.

Moreover, new information obtained from experimental studies could be considered by governing organizations that are involved in the regulation of environmental contaminants.

## Figures and Tables

**Figure 1 ijerph-16-00448-f001:**
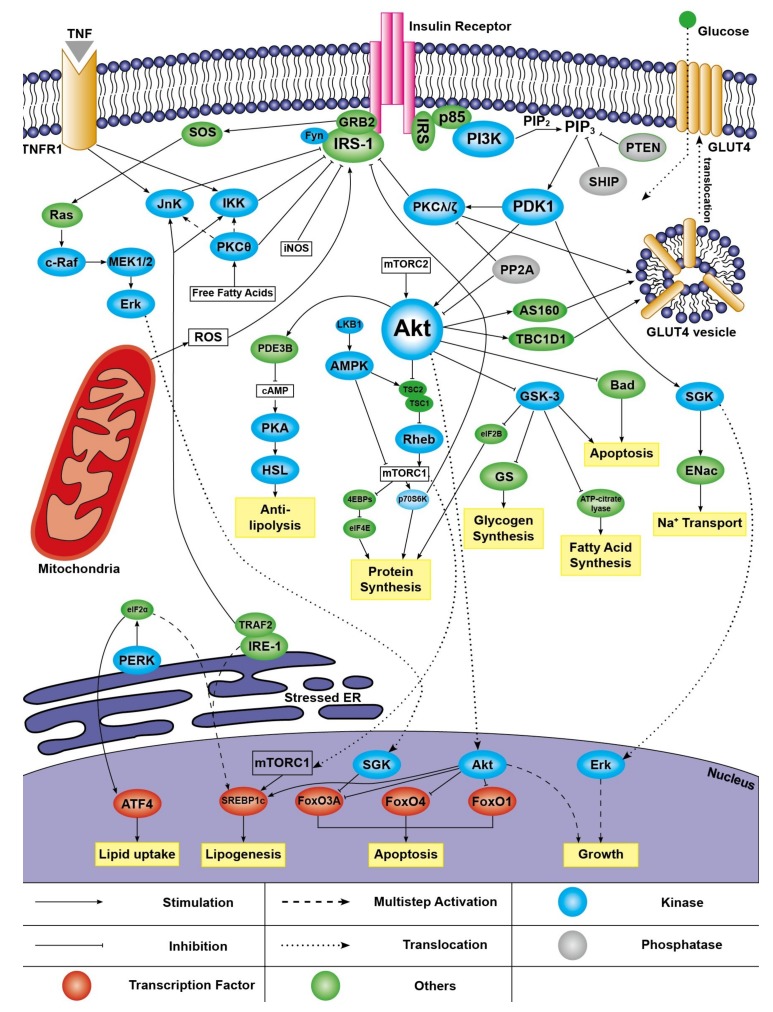
Current concepts of insulin signaling pathways.

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
