# Peer review of "Persistent Organic Pollutant-Mediated Insulin Resistance"

_ijerph, 2019, doi:10.3390/ijerph16030448_

Round 1

Reviewer 1 Report

Article: POPs-mediated insulin resistance

Authors: Kim YA, Park JB, Woo MS, Lee SY, Kim HY, Yoo H.

The subject area of the manuscript is of clear relevance and appropriate to the Journal. The manuscript is readily understandable and could be of interest for experts and non-experts. The description of the literature data is elaborated and well documented (111 references).

Comment: Please check the reference section for minor errors (ex: ref 10, add 3rd Edition, 2011; ref 15 JNCI:Journal.. delete JNCI... )

Author Response

Having appreciated the critiques raised by this reviewer, the reference section was corrected. In addition to the references mentioned by this reviewer, I corrected another two references in which there were typographical errors in author names.

Reviewer 2 Report

The paper "POPs-Mediated Insulin Resistance" is a well-written short review about recent knowledge on association of POPs exposure with insuline resistance.

I suggest the authors could try to enlarge the Chapter 6 about mechanism underlying POP induced insulin resistance. In my opinion this is what the reader expects and what is advertised in the title of this work. The title of Chapter 6 is "Experimental animal studies investigating the mechanism underlying POP-induced insulin resistance". Title could be simplified just to

"Experimental studies investigating the mechanism underlying POP-induced insulin resistance" and in vitro (tissue culture studies) on human cells could be included.

I think it would be for sake of the reader to include a piece of graphics showing insulin signaling pathways described in Chapter 5.

Minor points:

Title: I don't think the abbreviation POPs is suitable for the title. Better in full.

Throughout all the manuscript, inconsistency in text formatting occurs, probably caused by copy-paste last moment changes before submission. (line 42, line 48, line 63, line 70, line 86, line 283, line 284, line 327).

Chapter titles 7 and 8 are in different size than the others.

Spelling errors: line 28: have provided
                 line 114: prostate
                 line 115: effects of PCBs
                 line 123: and ?
                 line 218: carboxykinase (PEPCK)
                 line 220: de novo in italics

                 line 204: phosphor-inositide-3-kinase --> phosphoinositide 3-kinase

Please carefully check if all abbreviations are explained in time of the first use!!!

Not a full list, just what I randomly revealed: There are abbreviations explained several times

as polychlorinated biphenyls (PCBs) - line 44, line 53, line 58, line 156, line 175;

hexachlorobenzene - line 58, line 103

There are abbreviations never explained as LD  --> line 297 lipid droplets (LDs), BPA (line 98), PP2A (line 230). OCPs line 103,line 155, line 161, line 165;

POPs line 172   

abstract line 24,25,27 and throughout the manuscript: diabetes
The term diabetes may also refer to diabetes insipidus. I suggest you specify it (diabetes mellitus type 2) and define an abbreviation. Later in the text the abbreviation DM is used without explanation.

line 38. In my opinion, the sentence "In particular, endocrine disruptors (ECDs) disturb the immune, reproductive, and nervous system in humans and animals." doesn't fit to this place. Better to be stated somehow in Chapter 2 or 3.

I do not understand what you mean with "the early 1900s", several times in the text. The decade from 1900 to 1909? Or the 20th century? My personal feeling is that in 1905, there was no concept of insulin resistance defined yet and the industrial production of POPs was not so high as a little bit later.

Author Response

Having appreciated the critiques raised by this reviewer, I revised manuscript.

I appreciate very much that this reviewer thoroughly reviewed my manuscript. According to the opinion by this reviewer, I newly incorporated Fig. 1, in which current concepts on insulin signalling pathways are depicted. Furthermore, the critiques on usage of abbreviations,  typographical errors, and inconsistant format and, the usage of a term diabetes mellitus were all addressed. Regarding the expressiion indicating time point,  "the early 1900s" was replaced with "the 20th century".

Because I revised ALL as this reviewer suggested, point-to-point responses do not seem to be required.

Sincerely,